# SEMANTIC FRAGMENT SIMILARITY REPRESENTATION LEARNING FOR INFORMATION RETRIEVAL

## ABSTRACT

We introduce Semantic Fragment Similarity (SFS), a novel similarity metric designed to enhance representation quality by partitioning embeddings into non-overlapping fragments, computing fragment level similarity, and aggregating these local scores. Conventional similarity metrics compute relevance using the global vector as a single unit. This process flattens and entangles multi-faceted semantic features and dilutes the fine-grained alignment signals crucial for accuracy. By inducing fragments to specialize in distinct semantic roles, SFS drives the substantial gains in retrieval performance across a wide range of models, tasks, and architectures when applied in both training and inference. Further, we find that a single embedding fragment trained with SFS, comprising just 12% of the total dimensions, outperforms the entire global embedding on specific classification tasks. Ultimately, SFS can be directly integrated as a replacement for conventional similarity metrics, without architectural modifications or complex computational overhead and it opens up new avenues for building more structured and interpretable embedding models.

## 1 INTRODUCTION

As information retrieval advances, dense retrieval has become a widely used approach for efficiently retrieving the documents most relevant to a user query from large-scale collections (Karpukhin et al., 2020; Xiong et al., 2020). These approaches encode queries and documents as high-dimensional vectors and retrieve documents by their similarity in the embedding space. This representation captures complex semantic relationships beyond keyword matching (Khattab & Zaharia, 2020), leading to significant performance in applications such as retrieval-augmented generation (RAG) (Lewis et al., 2020; Gao et al., 2023; Jiang et al., 2023).

Dense retrieval models generate an embedding vector that encodes the overall meaning of a text into a single vector representation (Reimers & Gurevych, 2019; Li et al., 2020). The relevance of a query-document pair is quantified by a scalar score, typically derived from a similarity metric such as cosine similarity or the dot product computed between their vector representations (Günther et al., 2024; Lee et al., 2025b). This macro-level computation on the global vector effectively captures the overall meaning, but the process of reducing multifaceted semantics into a single scalar score inherently causes the flattening of local and multi-faceted object signals (Sinha et al., 2024). In this single-vector representation, different semantic features become entangled across the embedding's dimensions,

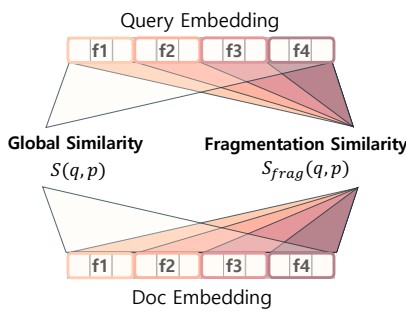

Figure 1: Conceptual Illustration of the Semantic Fragment Similarity.

consequently causing these distinct signals to interfere with or neutralize one another (You, 2025). For instance, if a query and a document exhibit a pronounced correspondence on one semantic axis but are irrelevant on another, the global similarity score can dilute this partial alignment, failing to capture the corresponding micro-level alignment in the vector space.

To mitigate representation flattening and entanglement, we propose Semantic Fragment Similarity (SFS), a novel metric that enhances the measurement of textual relevance by preserving local and

multi-faceted semantic signals. To encourage embeddings to encode distinct semantic aspects in different subspaces, SFS divides a high-dimensional vector into several non-overlapping fragments. It then calculates similarity for each query-document fragment pair and aggregates these local scores into a final similarity score. This approach allows SFS to explicitly retain fragment-level information, enabling a more fine-grained assessment of textual relatedness while still capturing global semantic context.

To demonstrate the effectiveness of our methodology, we conduct a comprehensive set of experiments applying SFS with diverse fragment size configurations to both embedding model training and retrieval phase. In these experiments, we train five transformer encoder models on retrieval data and conduct extensive evaluations on retrieval tasks in the MTEB benchmark (Muennighoff et al., 2023). The results show that the proposed SFS consistently outperforms a conventional learning approach based on global-vector calculation, yielding substantial improvements in retrieval performance with an average increase of 1–3% over the baseline. Moreover, these gains are observed across diverse embedding tasks (e.g., reranking, classification, and semantic similarity), pooling strategies, and model architectures, as partitioning into fine-grained fragments preserves and exploits local and multidimensional semantic features that would otherwise be diluted and flattened in a global-vector representation. Our analysis of fragment-level classification performance reveals that SFS encourages fragments to assume distinct semantic roles. This leads to cases where a single fragment, constituting just 12% of the total dimensions, outperforms the full embedding on certain classification tasks.

In summary, our contributions are: (1) We propose Semantic Fragment Similarity (SFS), a novel metric that mitigates the flattening and entanglement of semantic signals by aggregating fragment-wise similarities. This method requires no changes to the existing model architecture, such as adding new layers, parameters, or complex computations. (2) We experimentally demonstrate that SFS achieves substantial performance gains in our primary task, retrieval, while also consistently outperforming conventional global similarity across other tasks such as classification and reranking. (3) Our analysis reveals that SFS induces feature disentanglement and a semantic division of labor among fragments. We identify this mechanism as the driving force behind the observed improvements, offering a new lens that opens up possibilities for the interpretability of the embedding space.

## 2 SEMANTIC FRAGMENT SIMILARITY

In this section, we first introduce the training methodology for dense retrieval and the notation used throughout this paper. We then describe the training and inference processes using our proposed Semantic Fragment Similarity in detail.

### 2.1 PRELIMINARIES

In a dense retrieval framework using an embedding model, the objective is to retrieve semantically relevant documents from a large corpus $\mathcal{D} = \{d_1, d_2, \ldots, d_N\}$ given a query $q$. An embedding model $\mathcal{M}$ encodes textual inputs into a high-dimensional vector space of a fixed dimension. The model's output is then processed through a pooling operation to produce a dense vector in $R^{d_{model}}$, where $d_{model}$ denotes the model's output embedding dimension. For a given query $q$ and a document $d \in \mathcal{D}$, their embeddings are generated as follows:

$$\mathbf{E}_q = \text{Pooling}(\mathcal{M}(q)), \quad \mathbf{E}_d = \text{Pooling}(\mathcal{M}(d)) \in \mathbb{R}^{d_{\text{model}}} \tag{1}$$

The semantic relevance between a query $q$ and a document $d$ is quantified by a score $S(q, d)$, which is typically defined as the cosine similarity between their embedding vectors $E_q$ and $E_d$. This similarity function aggregates information across all dimensions of the representations and returns a single scalar that reflects the overall semantic similarity between the query and the document.

$$S(q, d) = \frac{\mathbf{E}_q \cdot \mathbf{E}_d}{\|\mathbf{E}_q\| \, \|\mathbf{E}_d\|} \tag{2}$$

The embedding model is trained to optimize retrieval performance using a contrastive learning objective. The goal is to maximize similarity score $S(q, d^+)$ for relevant query–positive document

pairs $(q, d^+)$ and minimize the similarity score $S(q, d^-)$ for non-relevant query–negative document pairs $(q, d^-)$.

## 2.2 Fragment Representation Learning

Our method, Semantic Fragment Similarity, learns to produce a final score by aggregating similarity scores calculated for each pair of corresponding fragments. These fragments are obtained by independently assessing the subspace correlation between two vectors within more granular subspaces, each representing a specific semantic aspect.

**Fine-grained Semantic Fragmentation** The objective of SFS is to produce a single scalar value that reflects micro-level alignment within the vector space. It does so by distinguishing strong signals confined to certain dimensions from weak signals dispersed across multiple dimensions.

We first define a fragmentation function $\mathcal{F}$ that partitions an embedding vector $E \in \mathbb{R}^{d_{\text{model}}}$ into an ordered sequence of non-overlapping fragments. Given a fragment dimension $d_{\text{frag}}$, the total number of fragments $N_f$ is determined as $d_{\text{model}}/d_{\text{frag}}$. The fragmentation function then maps the vector $E$ to a fragment sequence $\mathcal{F}(E) = \{\mathbf{f}_1, \mathbf{f}_2, \ldots, \mathbf{f}_{N_f}\}$, where each fragment $\mathbf{f}_i$ has a fixed dimension $\mathbb{R}^{d_{\text{frag}}}$. The fragment size $d_{\text{frag}}$, specified externally as a hyperparameter, ensures that all fragments have identical dimensionality. For instance, if the embedding dimension $d_{\text{model}} = 768$ and the fragment dimension $d_{\text{frag}} = 64$, the fragmentation function produces a sequence of 12 fragments $\{\mathbf{f}_1, \mathbf{f}_2, \ldots, \mathbf{f}_{12}\}$, each of dimension 64. Concretely, the first fragment $\mathbf{f}_1$ comprises the components of $E$ from indices 0 to 63. More generally, the $i$-th fragment $\mathbf{f}_i$ corresponds to the slice $E[(i-1) \times d_{frag} : i \times d_{frag}]$.

**Fragment-wise Similarity Aggregation** The query embedding $E_q$ and the document embedding $E_d$ are each partitioned into a sequence of $N_f$ fragments, $(\mathbf{f}_{q1}, \mathbf{f}_{q2}, \ldots, \mathbf{f}_{qN_f})$ and $(\mathbf{f}_{d1}, \mathbf{f}_{d2}, \ldots, \mathbf{f}_{dN_f})$, respectively. Instead of a global operation, we independently compute the cosine similarity, $s_i(q, d)$, for each corresponding fragment pair $(\mathbf{f}_{qi}, \mathbf{f}_{di})$ and then average the fragment-wise similarity scores to obtain a single similarity score $S_{\text{frag}}(q, d)$.

$$S_{\text{frag}}(q, d) = \frac{1}{N_f} \sum_{i=1}^{N_f} s_i(q, d) = \frac{1}{N_f} \sum_{i=1}^{N_f} \cos(\mathbf{f}_{qi}, \mathbf{f}_{di}) \tag{3}$$

This aggregation scheme equally weights alignment across semantic aspects, thereby preventing the flattening of information that arises during encoding and resulting in a final score that reflects micro-level alignment across the entire embedding space.

## 2.3 Training and Inference

**Training** To apply our proposed SFS in training, we integrate it into a contrastive objective. Specifically, within the infoNCE loss, we replace the conventional similarity function $S(q, d)$ with our $S_{\text{frag}}(q, d)$. The training objective is to minimize the following loss function:

$$L_{\text{NCE}} = -\frac{1}{n} \sum_i \log \frac{\exp(S_{\text{frag}}(q_i, d_i^+))}{\exp(S_{\text{frag}}(q_i, d_i^+)) + \sum_j \exp(S_{\text{frag}}(q_i, d_j^-))} \tag{4}$$

By minimizing this loss, the model learns to maximize fragment level semantic alignment for positive pairs $(q, d^+)$ and minimize it for negative pairs $(q, d^-)$. This training objective induces the model to structure the entire embedding space into a set of semantically coherent subspaces. Instead of relying on a single, global representation, the model learns to encode distinct semantic facets (e.g., topic, domain, and style) into different fragments of the embedding vector.

Consequently, each fragment becomes more adept at comparing these subtle and local semantic features, which leads to a more robust and interpretable similarity measure compared to a conventional single scalar value.

**Inference** At inference time, we encode the query and all documents into their respective embeddings. Subsequently, the final document ranking is determined using the Semantic Fragment Similarity $S_{frag}$. This score is computed by decomposing each query-document embedding pair into fragments, consistent with the fragment size defined during training, and then aggregating their local similarities. Algorithm 1 summarizes the specific inference procedure for ranking the entire document corpus for a given query.

---

**Algorithm 1** Inference with SFS

**Input:** Query embedding $\mathbf{E}_q$, Document embeddings $\{\mathbf{E}_{d_j}\}_{j=1}^N$, Fragment size $d_{\text{frag}}$
**Output:** Ranked documents
$N_f \leftarrow d_{\text{model}}/d_{\text{frag}}$
$\{\mathbf{f}_{qi}\}_{i=1}^{N_f} \leftarrow \mathcal{F}(\mathbf{E}_q)$
**for** $j \leftarrow 1$ **to** $N$ **do**
  $\{\mathbf{f}_{d_j i}\}_{i=1}^{N_f} \leftarrow \mathcal{F}(\mathbf{E}_{d_j})$
  $S_{\text{frag}}(\mathbf{E}_q, \mathbf{E}_{d_j}) \leftarrow \frac{1}{N_f} \sum_{i=1}^{N_f} \cos(\mathbf{f}_{qi}, \mathbf{f}_{d_j i})$
**end**
**return** sort(Documents, by=$S_{\text{frag}}$)

---

## 3 EXPERIMENTAL SETUP

**Training** Our primary experiments focus on transformer encoder architectures. We select five models, all with a 768-dimension hidden state: ModernBERT-base (Warner et al., 2025), bert-base-uncased (Devlin et al., 2019), gte-en-mlm-base (Zhang et al., 2024), roberta-base (Liu et al., 2019), and nomic-bert-2048 (Nussbaum et al., 2025). We employ the [CLS] pooling strategy to generate sentence embeddings. For each fragment size in $d_{\text{frag}} \in \{768, 256, 128, 64, 32, 16\}$, we train a dedicated model and subsequently evaluate it under the same size configuration. In all experiments, configurations are denoted as (number of fragments, fragment size).

We utilize the standard InfoNCE (van den Oord et al., 2018) loss over in-batch negatives and hard negatives (Henderson et al., 2017). For training data, We employ a collection of publicly available datasets that are standard training datasets in the embedding literature. Further details regarding our overall implementation and the training datasets are provided in Appendix B.

**Evaluation** We conduct evaluations on the Massive Text Embedding Benchmark (MTEB) (Muennighoff et al., 2023). Our experiments focus on a curated subset of 31 datasets from four task categories: retrieval, reranking, classification, and sentence similarity (STS). All evaluations are conducted using the official MTEB to ensure consistency and reproducibility. A comprehensive list of the datasets included in our evaluation is available in Appendix C

## 4 EXPERIMENTAL RESULTS

### 4.1 RETRIEVAL

Table 1: Retrieval performance (nDCG@10) by fragment size. Configurations are denoted as (Number of Fragments, Fragment Size). **Bold** and underlined values are the best and second-best scores for each model, respectively.

| **Model** | (1,768) | (3,256) | (6,128) | (12,64) | (24,32) | (48,16) |
|-----------|---------|---------|---------|---------|---------|---------|
| BERT | 0.5178 | 0.5136 | 0.5185 | 0.5163 | **0.5197** | 0.5181 |
| RoBERTa | 0.5045 | 0.5082 | 0.5099 | 0.5186 | 0.5168 | **0.5195** |
| NomicBERT | 0.5356 | 0.5346 | 0.5388 | 0.5414 | 0.5414 | **0.5427** |
| ModernBERT | 0.5375 | 0.5345 | 0.5330 | 0.5257 | 0.5483 | **0.5512** |
| GTE-en-MLM | 0.5424 | 0.5437 | 0.5398 | 0.5500 | **0.5517** | 0.5508 |

Table 1 presents retrieval performance across five models with varying fragment sizes. It shows that our proposed Semantic Fragment Similarity consistently outperforms similarity computations based on global vectors. Across all models evaluated in the experiments, partitioning the vectors into multiple fragments to compute similarity yields better results than the conventional single-fragment setting of (1,768).

Notably, all five models used in the experiments tend to achieve higher performance as the embeddings are segmented into more granular fragments, specifically when the fragment size decreases to 16 or 32. For example, the ModernBERT demonstrated the most significant improvement, achieving approximately a 2.5% gain over the baseline when the fragment size was set to 16. These findings

Table 2: Performance comparison on Reranking, Classification, and STS tasks.

| Model | (1,768) | (3,256) | (6,128) | (12,64) | (24,32) | (48,16) |
|---|---|---|---|---|---|---|
| | | | *Reranking* | | | |
| BERT | 0.4972 | 0.4989 | 0.4988 | 0.4984 | 0.5004 | **0.5007** |
| RoBERTa | 0.4987 | 0.4986 | 0.4997 | 0.4975 | 0.4996 | **0.5000** |
| NomicBERT | 0.4972 | **0.4974** | 0.4950 | 0.4960 | 0.4961 | 0.4969 |
| ModernBERT | 0.5260 | 0.5269 | 0.5267 | 0.5260 | 0.5283 | **0.5303** |
| GTE-en-MLM | 0.5173 | 0.5163 | 0.5150 | 0.5160 | **0.5178** | 0.5173 |
| | | | *Classification* | | | |
| BERT | 0.5444 | 0.5448 | 0.5448 | 0.5476 | 0.5479 | **0.5487** |
| RoBERTa | 0.5585 | 0.5570 | 0.5620 | 0.5699 | **0.5725** | 0.5677 |
| NomicBERT | 0.5320 | 0.5325 | 0.5333 | 0.5420 | **0.5439** | 0.5427 |
| ModernBERT | 0.5483 | 0.5451 | 0.5470 | 0.5507 | 0.5586 | **0.5613** |
| GTE-en-MLM | 0.5453 | 0.5450 | 0.5490 | 0.5490 | **0.5551** | 0.5502 |
| | | | *Semantic Textual Similarity* | | | |
| BERT | 0.7350 | 0.7364 | 0.7389 | 0.7384 | 0.7416 | **0.7428** |
| RoBERTa | 0.7330 | **0.7342** | 0.7305 | 0.7313 | 0.7337 | 0.7341 |
| NomicBERT | 0.7434 | 0.7421 | 0.7367 | 0.7373 | 0.7413 | **0.7439** |
| ModernBERT | 0.7509 | 0.7490 | **0.7520** | 0.7456 | 0.7485 | 0.7470 |
| GTE-en-MLM | 0.7497 | 0.7514 | 0.7476 | 0.7516 | 0.7532 | **0.7549** |

indicate that our approach partitions the high dimensional vector space into multiple fine grained semantic subspaces, evaluates local similarities within each subspace independently, and subsequently aggregates the resulting similarity measures, enabling more effective capture of the multifaceted and subtle relationships.

## 4.2 Evaluation on Non-Retrieval Embedding Tasks

Table 2 presents the performance variations in reranking, classification, and STS tasks across different fragment sizes, using the full (1,768) embedding as the baseline. Performance on reranking and classification tasks steadily improves as fragments become more granular; in classification, ModernBERT and RoBERTa achieve performance gains of 2.3% and 2.5%, respectively. For STS, while the optimal fragment size varies across models, more fine-grained fragmentation generally yields better results. This suggests that semantic fragmentation enables embeddings to effectively represent local semantic information and provides a robust approach to improving performance across diverse downstream tasks.

## 5 How Robust Is Fragment Representation Learning in Retrieval?

In this section, we conduct in-depth experiments to assess whether representation learning with the proposed Fragment Similarity remains effective under varying conditions in retrieval tasks. Specifically, we systematically and comprehensively explore variations in pooling strategy, fragment-size scaling, model architecture, and similarity metrics.

### 5.1 Effectiveness Across Different Pooling Methods

**Setup** To assess robustness across pooling strategies, we replace [CLS] pooling with average pooling. Average pooling generates embeddings by averaging all token representations from the last hidden outputs. We train two models, ModernBERT and BERT, under the same experimental settings as the [CLS] configuration, except for the pooling strategy.

**Results** As shown in Table 3, overall performance with average pooling is slightly lower or on par with that obtained [CLS] pooling. Crucially, we observe consistent gains over the full-embedding baseline (1×768). For example, ModernBERT-base improves from 0.5390 to as high as 0.5444, consistent with the improvement trend observed under [CLS] pool-

Table 3: Performance comparison of different fragment sizes with average pooling.

| (1,768) | (3,256) | (6,128) | (12,64) | (24,32) | (48,16) |
|---|---|---|---|---|---|
| | | **BERT** | | | |
| 0.5215 | 0.5228 | 0.5203 | **0.5241** | 0.5211 | 0.5232 |
| | | **ModernBERT** | | | |
| 0.5390 | 0.5403 | 0.5439 | 0.5323 | **0.5444** | 0.5417 |

Table 4: The effect of fragment granularity scaling on retrieval performance.

| Model | (1,768) | (2,384) | (3,256) | (4,192) | (6,128) | (12,64) | (24,32) | (48,16) | (96,8) | (192,4) | (384,2) |
|---|---|---|---|---|---|---|---|---|---|---|---|
| BERT | 0.5178 | 0.5191 | 0.5136 | 0.5175 | 0.5185 | 0.5163 | **0.5197** | 0.5181 | 0.5192 | 0.5144 | 0.5053 |
| ModernBERT | 0.5375 | 0.5370 | 0.5345 | 0.5368 | 0.5330 | 0.5257 | 0.5483 | **0.5512** | 0.5502 | 0.5375 | 0.5242 |

ing. This suggests that average pooling can, via fragment-wise consensus across subspaces, mitigate the influence of outlier token representations and, in turn, yield more robust representations.

## 5.2 FRAGMENT GRANULARITY SCALING

**Setup** To more thoroughly investigate the impact of fragment granularity, we evaluate an expanded range of fragment configurations on BERT and ModernBERT. In addition to those previously evaluated, we include intermediate granularities ($d_{\text{frag}} \in \{384, 192\}$) and more fine-grained settings ($d_{\text{frag}} \in \{8, 4, 2\}$).

**Results** Table 4 reveal a clear trend with respect to fragment granularity. A consistent pattern is observed where performance improves as fragments become more granular, peaks at a certain point, and then sharply declines when the fragmentation becomes extreme. Generally, optimal performance is achieved within the more granular range of fragment sizes, specifically around 32, 16, and 8. This pattern suggests a fundamental trade-off related to the role of fragments. Fragments must be small enough to encode distinct semantic aspects, yet large enough to retain the minimum dimensional capacity required to encode complex meanings. Extreme fragmentation undermines this capacity, leading to performance degradation. Therefore, These findings suggest that an optimal level of granularity exists to maximize the effectiveness of SFS.

## 5.3 ROBUSTNESS ACROSS MODEL ARCHITECTURES

Table 5: Retrieval performance of different fragmentation configurations on decoder models.

| Model | (1,2048) | (2,1024) | (4,512) | (8,256) | (16,128) | (32,64) | (64,32) | (128,16) |
|---|---|---|---|---|---|---|---|---|
| Qwen3-0.6B | 0.5762 | 0.5799 | 0.5848 | 0.5794 | 0.5842 | **0.5888** | 0.5847 | 0.5874 |
| Llama-3.2-1B | 0.5939 | 0.6018 | 0.6002 | 0.6014 | 0.5953 | 0.5983 | 0.5980 | **0.6028** |

**Setup** We evaluate our method on decoder architectures by training two models, Qwen3-0.6B Team (2025) and Llama-3.2-1B, which have a hidden dimension of 2048. We utilize LLMs with causal attention, appending an [EOS] token at the end of the input sequence (Zhang et al., 2025). The sentence embedding is derived from the hidden state of the last layer corresponding to this [EOS] token.

**Results** Table 5 shows a common trend that learning via fragment similarity yields consistent performance improvements over the baseline for decoder architectures. Both models significantly outperform the baseline with more granular fragmentation, with Qwen3-0.6B achieving a top score of 0.5888 ($d_{\text{frag}} = 64$) and Llama-3.2-1B a top score of 0.6028 ($d_{\text{frag}} = 16$). These results suggest that, similar to encoder models, the embedding space of decoder models is also composed of mixed, functionally distinct features. Our approach effectively separates these antagonistic signals to enhance local alignment, resulting in superior representational quality.

## 5.4 EFFECTIVENESS ACROSS SIMILARITY METRICS

**Setup** We conducted an experiment to determine if the benefits of fragment-based learning generalize beyond cosine similarity. Using the ModernBERT-base model, we compared a full-embedding baseline (1,768) against a fragmented configuration (48,16) across four metrics: Cosine, Euclidean (L2), Manhattan (L1), and Dot product.

Table 7: Classification performance across different fragmentation configurations on the Toxic Conversation and Amazon Counterfactual datasets.

| Dataset | (1, 768) | (2, 384) | (3, 256) | (4, 192) | (6, 128) | (12, 64) | (24, 32) | (48, 16) |
|---|---|---|---|---|---|---|---|---|
| Toxic Conversation | 0.4737 | 0.4743 | 0.4807 | 0.4799 | 0.4809 | 0.5165 | 0.5126 | **0.5281** |
| Amazon Counterfactual | 0.5296 | 0.5305 | 0.5333 | 0.5336 | 0.5371 | 0.5468 | **0.5694** | 0.5686 |

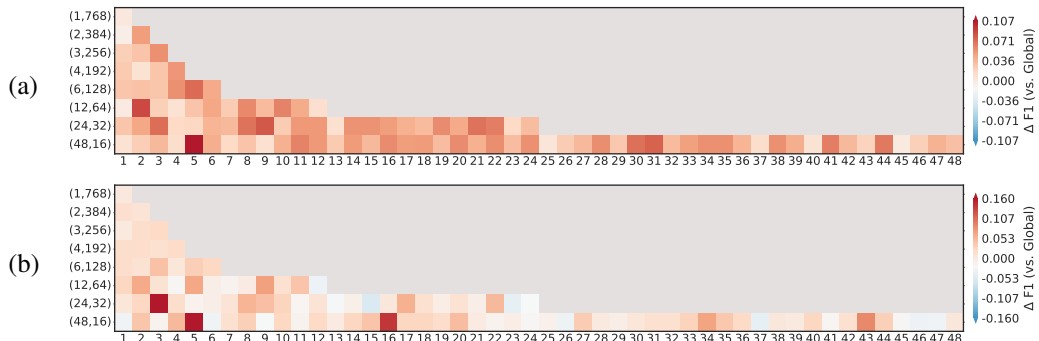

Figure 2: Visualization of single-fragment classification performance relative to the global embedding. Results are shown for (a) the Toxic Conversation and (b) the Amazon Counterfactual.

**Results** Table 6 demonstrates that our Fragment Similarity-based learning is effective for similarity metrics in addition to cosine. Applying fragmentation improves performance across all metrics. Notably, the improvements are substantially larger for metrics sensitive to vector magnitude, such as Euclidean (L2), Manhattan (L1), and the dot product. These results suggest that our method effectively mitigates the problem of anisotropy in the embedding space. The process of measuring and aggregating similarity at the fragment level reduces the directional bias and scale sensitivity of the full vector representation, which we identify as the key reason for the more significant performance enhancements in these metrics.

Table 6: Performance comparison of different similarity metrics, with and without Fragment Similarity.

| Metric | (1,768) | (48,16) |
|---|---|---|
| Cosine | 0.5375 | **0.5512** |
| Euclidean | 0.3348 | **0.4680** |
| Manhattan | 0.3582 | **0.5145** |
| Dot | 0.3463 | **0.5143** |

## 6 FRAGMENT-LEVEL ANALYSIS AND COMPARATIVE EVALUATION

### 6.1 DO FRAGMENTS LEARN DISTINCT SEMANTIC ROLES?

In this section, we experimentally analyze whether training with the proposed methodology indeed induces each fragment of the embedding to assume semantically differentiated roles (distinct semantic roles). Using a ModernBERT-base model trained under various fragment configurations, we conduct two complementary experiments. In the first experiment, we directly compare the classification performance of the full embedding and individual fragments. In the second experiment, we statistically analyze, from an in-depth perspective, how specific fragments respond to specific semantic attributes (semantic features).

### 6.1.1 VALIDATING SPECIALIZATION VIA CLASSIFICATION PERFORMANCE

**Setup** We evaluate on two classification datasets with clearly delineated topics, Toxic Conversation and Amazon Counterfactual. We first measure performance using the full embedding, and then assess the performance contribution of individual fragments by comparing it to the performance obtained when using each fragment vector alone.

**Results** Table 7 shows that as fragment partitioning becomes more fine-grained, classification performance improves consistently. Relative to the baseline, Toxic Conversation increases by up to 11.48%, and Amazon Counterfactual by up to 7.51%. To probe whether the improvements are

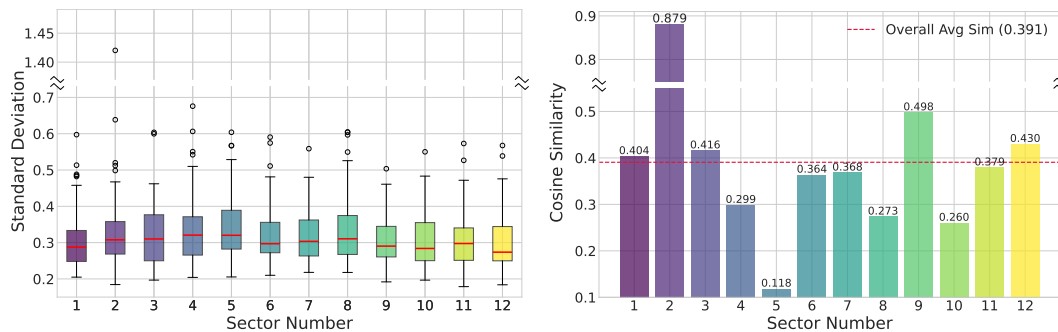

(a) Standard Deviation Distribution per Fragment.
(b) Average Cosine Similarity per Fragment.

Figure 3: **Analysis of fragment-level responses to the semantic attribute of toxicity.** (a) displays the distribution of dimension-wise standard deviations for the embedding difference vectors between toxic/neutral sentence pairs, grouped by fragment. (b) shows the average cosine similarity for each fragment between the sentence pairs.

driven by overall embedding quality or by the semantic specialization of individual fragments, Figure 2 compares the performance of each fragment with that of the full vector. As the number of partitions increases, some fragments clearly outperform the full vector; in particular, in the Toxic Conversation (12,64) model, several fragments surpass the whole, and the 5th fragment exceeds it by more than 0.1. In other words, using only 16 dimensions (about 12% of 768) can outperform the full embedding, suggesting that training with fragment similarity concentrates semantic signals within fragments.

### 6.1.2 ANALYZING THE SPECIALIZATION MECHANISM

**Setup** Building on the previous analysis, we statistically demonstrate how specific fragments encode semantic attributes. In this experiment, we use a model trained with 12 fragments of 64 dimensions each and the TextDetox Dementieva et al. (2025) dataset containing 400 Toxic/Neutral sentence pairs. To quantify fragment-wise activation differences, we compute the dimension-wise standard deviation of the embedding difference vectors between sentence pairs, group these statistics by fragment and analyze their distributions; in parallel, we measure the average cosine similarity between the fragment embeddings of the sentence pairs.

**Results** The most notable finding is observed in the second fragment. While this fragment exhibits the highest average cosine similarity across toxic/neutral sentence pairs (Figure 3(b)), it concurrently achieves superior toxicity classification performance for the (12,64) model, as depicted in Figure 2(a). This result is attributable to the characteristics shown in Figure 3(a): a small subset of dimensions within this fragment presents substantial standard deviations in the pairwise difference vectors. This indicates that a minority of dimensions robustly encodes the distinction between toxic and neutral representations, while the majority remain largely invariant. Collectively, these observations validate that our methodology induces fragments to specialize in encoding distinct semantic attributes.

### 6.2 COMPARISON WITH MATRYOSHKA REPRESENTATION LEARNING

**Setup** We compare the efficiency of representation with Matryoshka Representation Learning (MRL) Kusupati et al. (2022), a representative compression approach for adaptive-dimensional embeddings. MRL concentrates information importance in the prefix of the embedding so that strong performance can be achieved with only a subset of dimensions. For MRL, we train ModernBERT at embedding sizes [768, 512, 384, 256, 128, 64]. For our method, we use a model trained with 48 fragments of 16 dimensions each. For a fair comparison at the same dimensionality, we construct our representation by concatenating fragments of size 16 in order until the target dimension is reached. For example, the 256 dimensional variant uses the first 16 fragments and is compared directly to MRL at 256 dimensions.

**Results** Figure 4 presents a comparison of retrieval performance across representation sizes. Both methods show a common pattern of improvement as dimensionality increases. For dimensionalities at or below 128, MRL holds a slight advantage, which we attribute to its training objective that concentrates critical information in the prefix of the embedding. However, after the curves cross at 256 dimensions, our method consistently outperforms MRL at all larger sizes. In particular, at 512 dimensions we observe approximately a 4.34% gain in nDCG@10, and the gap continues to widen as dimensionality increases. These results suggest that, unlike MRL, which compresses information into a subset of dimensions, our approach regards each

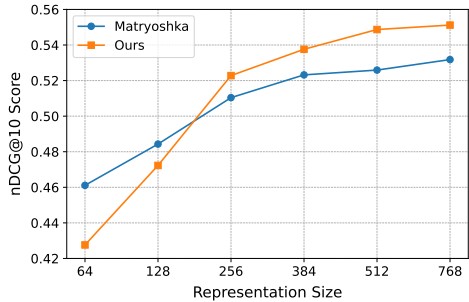

Figure 4: Retrieval Performance Comparison with Matryoshka Representation Learning

fragment as an independent semantic unit and leverages complementary interactions across fragments to improve representational quality. In addition, these findings suggest that fragment training based on fragment similarity can also serve an adaptive dimensionality.

## 7 RELATED WORK

One factor that undermines the expressiveness of embedding models and degrades retrieval performance is representation dilution. A major cause of representation dilution arises in the pooling process that aggregates token-level representations into a single document vector. Pooling inevitably incurs information loss as it compresses the rich semantic content of the input text into a single vector. In this regard, Lee et al. (2025a) point out that mean pooling dilutes information about salient spans, and that [EOS] pooling can suffer from recency bias (Lin et al., 2025), thereby clarifying the limitations of both approaches. This dilution problem becomes more severe as text length increases: Hu et al. (2025) note that representing a long context with a single vector can lead to representation collapse (Zhou et al., 2024) or dilution. Furthermore, structural characteristics of the model, such as causal attention, can also induce dilution of discriminative information in the latter part of the text (Springer et al., 2024).

Beyond these characteristics, the similarity metric itself also acts as a key contributing factor. In practice, many recent embedding models adopt cosine similarity as the primary similarity measure (Günther et al., 2024; Lee et al., 2025b; Muennighoff et al., 2024; Choi et al., 2024). In particular, cosine similarity is directly included in the training objective of contrastive learning, such as the InfoNCE loss, playing a decisive role in shaping the model's representation space. However, cosine similarity has a fundamental limitation: it gives rise to anisotropy in the embedding space (You, 2025; Liang et al., 2021). This issue causes most vectors to cluster within a narrow cone region in high-dimensional space, undermining semantic discriminability, and inducing a distance concentration effect where pairwise distances become similar, thus degrading the reliability of retrieval systems (Wang & Isola, 2020). Additionally, other metrics such as Euclidean distance can flatten complex hierarchical semantic relations, distorting the original semantic context (Sinha et al., 2024).

## 8 CONCLUSION

In this paper, we propose Semantic Fragment Similarity (SFS) to address how global similarity calculations in Dense Retrieval flatten multifaceted semantic information, which degrades representation quality. SFS operates by partitioning an embedding vector into multiple non-overlapping fragments, independently computing similarity at the fragment level, and aggregating these local scores. This approach induces the model to assign distinct semantic roles to each fragment, and our experiments demonstrate that it consistently outperforms conventional global similarity calculation across a diverse range of models and tasks. Notably, this approach induces a semantic division of labor, evidenced by our analysis where a single fragment can outperform the entire embedding on specific tasks. Ultimately, SFS challenges the conventional paradigm of relying on global-level operations, presenting a new direction for more structured and reliable models, though future work remains in areas like optimal partitioning and the automatic identification of fragment roles.

## ETHICS STATEMENT

This research raises no direct ethical concerns. It does not involve human participants, personal or sensitive information, or animal experiments. All datasets are publicly available and legally licensed for research use. The models and methodologies employed adhere to community-accepted ethical practices and are not designed for harmful, malicious, or discriminatory purposes. We believe this work aligns with responsible AI research principles and does not introduce foreseeable risks of misuse.

## REPRODUCIBILITY STATEMENT

Our research is designed for full reproducibility. All datasets, models, and training methodologies are described in detail to allow for easy replication. The experimental setup is thoroughly documented in Section 3, with specific training details, including hyperparameters and computational resources used, provided in Appendix B. We also provide a complete list of benchmark datasets used for evaluation in Appendix C.

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

# A  LIMITATIONS

While the proposed Semantic Fragment Similarity (SFS) improves retrieval performance, several limitations highlight important directions for future work. First, although we have experimentally confirmed that SFS training induces a semantic division of labor among fragments, intuitively interpreting and naming the specific meaning that each fragment specializes in remains a challenge. The absence of a systematic method to quantify the semantic function of each fragment is a factor that limits the model's interpretability. This limitation in interpretability is also directly linked to another assumption of our study: that SFS computes the final score by simply averaging the similarity scores of the fragments. This approach carries the implicit premise that all fragments contribute equally to relevance judgments for any given task. If future research enables the identification and quantification of each fragment's semantic function, it would be possible to devise a more sophisticated aggregation mechanism that moves beyond simple averaging to dynamically assign weights tailored to the specific characteristics of a task.

# B  TRAINING DETAILS

## B.1  SETUP

All training is conducted on four NVIDIA A6000 GPUs. We employ GradCache (Gao et al., 2021) to train with a large batch size (512) with limited GPU memory. Models are trained for 2 epochs with a maximum sequence length of 512, using bf16 precision. We use the AdamW optimizer (Loshchilov & Hutter, 2019) with a learning rate of 1e-4 and a linear learning rate warm-up for 5% of the total steps. Additionally, to ensure reproducibility, all random seeds are fixed.

Table 8: Instructions and number of samples used for each training dataset.

| Task Name | Instruction | # of samples |
|---|---|---|
| arguana | Given a claim, retrieve documents that support or refute the claim | 4.1k |
| fever | Given a claim, retrieve documents that support or refute the claim | 29.1k |
| scifact | Given a scientific claim, retrieve documents that support or refute the claim | 0.5k |
| paq | Given a web search query, retrieve relevant passages that answer the query | 45.2k |
| msmarco_document | Given a question, retrieve documents that can help answer the question | 50k |
| msmarco_passage | Given a question, retrieve passages that can help answer the question | 50k |
| squad | Given a question, retrieve passages that answer the question | 87.6k |
| natural question | Given a question, retrieve passages that answer the question | 58.6k |
| hotpotqa | Given a multi-hop question, retrieve documents that can help answer the question | 84.5k |
| fiqa | Given a financial question, retrieve relevant passages that answer the query | 5.5k |
| miracl | Given a question, retrieve passages that answer the question | 7.9k |
| mrtydi | Given a question, retrieve passages that answer the question | 3.5k |
| gooaq | Given a question, retrieve passages that answer the question | 40.4k |
| eli5 | Given a question, retrieve passages that answer the question | 21k |
| trivia_qa | Given a question, retrieve passages that answer the question | 31.7k |

## B.2  DATASET

We leverage the training data provided by BGE-en-ICL (Li et al., 2024) as follows (Lei et al., 2025) along with a collection of publicly available retrieval datasets. We adopt the retrieval datasets as follows: ArguAna (Wachsmuth et al., 2018), FEVER (Thorne et al., 2018), SciFact (Wadden et al., 2020), PAQ (Lewis et al., 2021), MSMARCO (Bajaj et al., 2016), SQuAD (Rajpurkar et al., 2016), Natural Question (Kwiatkowski et al., 2019), HotpotQA (Yang et al., 2018), FiQA (Maia et al., 2018), MIRACL (Zhang et al., 2023), Mr.TyDi (Zhang et al., 2021), GooAQ (Khashabi et al., 2021), ELI5 (Fan et al., 2019) and TriviaQA (Joshi et al., 2017). The full list of detailed datasets and their corresponding instructions is provided in Table 8.

**Hard Negative Mining**  To construct a high-quality training dataset, we employ a hard negative mining process (de Souza P. Moreira et al., 2024). This process begins by utilizing bge-large-en-v1.5 (Xiao et al., 2023) to convert all queries and passages into dense vector representations. For each query, we perform an efficient, corpus-wide semantic search to rank all passages by similarity. From this ranked list, we select effective hard negatives. We adopt a top sampling strategy and apply an absolute margin of 0.04. From the resulting candidates, we sample 7 hard negatives for each sample.

# C EVALUATION DETAILS

Table 9: Subset of MTEB tasks and benchmarks used for our experiments.

| Task Category | Benchmarks |
| --- | --- |
| Retrieval (12) | NanoArguAna, NanoClimateFEVER, NanoDBPEDIA, NanoFEVER, NanoFiQA2018, NanoHotpotQA, NanoMSMARCO, NanoNFCorpus, NanoQuoraRetrieval, NanoSCIDOCS, NanoSciFact, NanoTouche2020 |
| Reranking (4) | AskUbuntuDupQuestions, MindSmallRerank, SciDocsRR, StackOverflowDupQuestions |
| Classification (7) | AmazonCounterfactual, AmazonPolarity, AmazonReviews, Banking77, Imdb, ToxicConversations, TweetSentimentExtraction |
| STS (8) | BIOSSES, STS12, STS13, STS14, STS15, STS16, STS17, STS22 |

Table 9 lists all the evaluation tasks. We evaluate all five fine-tuned models on the MTEB Benchmark encompassing 12 retrieval datasets, 4 reranking datasets, 7 classification datasets, and 8 semantic textual similarity datasets. Due to the significant computational cost of the full benchmark, we specifically utilize NanoBEIR Thakur et al. (2021) for the retrieval tasks (Takeshita et al., 2025). For the classification, we include datasets that are evaluated using the k-NN classification. Furthermore, to align with our model's learning mechanism, similarity is specifically measured using our proposed fragmentation similarity. For evaluation, we use the prompts provided by the library [1] for each benchmark.

# D DETAILED EXPERIMENTAL RESULTS

This appendix provides a comprehensive breakdown of the performance metrics for each model and component analysis evaluated in our study.

First, we present the detailed results for the main experiments. The performance breakdown for `ModernBERT` is presented in Table 13, followed by the results for `BERT` in Table 10, `GTE-en-MLM` in Table 14, `NomicBERT` in Table 12, and `RoBERTa` in Table 11.

Additionally, we provide the results from our ablation and component analysis studies. Table 15 details the impact of different **pooling methods**, while the effects of **representation scaling** are shown in Table 16. The results related to the **decoder architecture** and the choice of **similarity metric** are documented in Table 17 and Table 18, respectively. Finally, the analysis of the **Matryoshka Representation Learning (MRL)** training objective is provided in Table 19.

---

[1] https://github.com/embeddings-benchmark/mteb

Table 10: Full Benchmark Results for BERT

| Dataset | (1,768) | (3,256) | (6,128) | (12,64) | (24,32) | (48,16) |
|---|---|---|---|---|---|---|
| ArguAna | 0.5780 | 0.5757 | 0.5873 | 0.5837 | 0.5963 | 0.5736 |
| ClimateFEVER | 0.2224 | 0.2273 | 0.2290 | 0.2243 | 0.2266 | 0.2181 |
| DBPEDIA | 0.5148 | 0.5129 | 0.5132 | 0.5089 | 0.5168 | 0.5155 |
| FEVER | 0.8221 | 0.8034 | 0.8208 | 0.8172 | 0.8253 | 0.8306 |
| FiQA2018 | 0.3847 | 0.3730 | 0.3810 | 0.3862 | 0.3788 | 0.3774 |
| HotpotQA | 0.6313 | 0.6336 | 0.6364 | 0.6222 | 0.6263 | 0.6330 |
| MSMARCO | 0.5693 | 0.5595 | 0.5609 | 0.5503 | 0.5721 | 0.5661 |
| NFCorpus | 0.2542 | 0.2627 | 0.2628 | 0.2679 | 0.2646 | 0.2690 |
| QuoraRetrieval | 0.9267 | 0.9208 | 0.9210 | 0.9199 | 0.9193 | 0.9203 |
| SCIDOCS | 0.2475 | 0.2446 | 0.2437 | 0.2468 | 0.2458 | 0.2487 |
| SciFact | 0.5675 | 0.5644 | 0.5685 | 0.5648 | 0.5651 | 0.5571 |
| Touche2020 | 0.4695 | 0.4615 | 0.4688 | 0.4665 | 0.4679 | 0.4739 |
| AskUbuntuDupQuestions | 0.5502 | 0.5551 | 0.5533 | 0.5534 | 0.5585 | 0.5570 |
| SciDocsRR | 0.7021 | 0.7021 | 0.7023 | 0.7017 | 0.7037 | 0.7046 |
| StackOverflowDupQuestions | 0.4248 | 0.4266 | 0.4284 | 0.4271 | 0.4279 | 0.4302 |
| MindSmallRerank | 0.3117 | 0.3117 | 0.3113 | 0.3113 | 0.3113 | 0.3111 |
| BIOSSES | 0.8170 | 0.8195 | 0.8196 | 0.8236 | 0.8218 | 0.8152 |
| STS12 | 0.6246 | 0.6268 | 0.6277 | 0.6242 | 0.6276 | 0.6296 |
| STS13 | 0.7893 | 0.7885 | 0.7917 | 0.7907 | 0.7932 | 0.7956 |
| STS14 | 0.6952 | 0.6969 | 0.6995 | 0.6996 | 0.7036 | 0.7056 |
| STS15 | 0.7836 | 0.7852 | 0.7878 | 0.7860 | 0.7894 | 0.7912 |
| STS16 | 0.7571 | 0.7572 | 0.7568 | 0.7579 | 0.7607 | 0.7625 |
| STS17 | 0.7689 | 0.7729 | 0.7810 | 0.7760 | 0.7836 | 0.7862 |
| STS22 | 0.6439 | 0.6444 | 0.6468 | 0.6490 | 0.6528 | 0.6568 |
| AmazonCounterfactual | 0.5509 | 0.5535 | 0.5569 | 0.5679 | 0.5773 | 0.5730 |
| AmazonPolarity | 0.5981 | 0.5971 | 0.5969 | 0.6002 | 0.6021 | 0.6034 |
| AmazonReviews | 0.2756 | 0.2751 | 0.2745 | 0.2742 | 0.2773 | 0.2777 |
| Banking77 | 0.6849 | 0.6859 | 0.6884 | 0.6869 | 0.6900 | 0.6902 |
| Imdb | 0.5725 | 0.5725 | 0.5707 | 0.5692 | 0.5761 | 0.5772 |
| ToxicConversations | 0.5027 | 0.5048 | 0.5090 | 0.5326 | 0.5099 | 0.5106 |
| TweetSentimentExtraction | 0.4285 | 0.4320 | 0.4293 | 0.4281 | 0.4316 | 0.4313 |

Table 11: Full Benchmark Results for RoBERTa

| Dataset | (1,768) | (3,256) | (6,128) | (12,64) | (24,32) | (48,16) |
|---|---|---|---|---|---|---|
| ArguAna | 0.6112 | 0.6030 | 0.6194 | 0.6198 | 0.6258 | 0.6256 |
| ClimateFEVER | 0.2098 | 0.2481 | 0.2348 | 0.2414 | 0.2225 | 0.2549 |
| DBPEDIA | 0.4890 | 0.4857 | 0.4929 | 0.4983 | 0.5017 | 0.5011 |
| FEVER | 0.6952 | 0.6969 | 0.6939 | 0.6800 | 0.6886 | 0.7053 |
| FiQA2018 | 0.4101 | 0.4284 | 0.4317 | 0.4432 | 0.4304 | 0.4436 |
| HotpotQA | 0.6102 | 0.6178 | 0.6209 | 0.6288 | 0.6540 | 0.6392 |
| MSMARCO | 0.5601 | 0.5590 | 0.5513 | 0.5835 | 0.5763 | 0.5569 |
| NFCorpus | 0.2357 | 0.2373 | 0.2292 | 0.2397 | 0.2380 | 0.2386 |
| QuoraRetrieval | 0.9415 | 0.9447 | 0.9361 | 0.9433 | 0.9260 | 0.9462 |
| SCIDOCS | 0.2720 | 0.2768 | 0.2797 | 0.2857 | 0.2774 | 0.2916 |
| SciFact | 0.5163 | 0.5001 | 0.4999 | 0.5263 | 0.5436 | 0.5147 |
| Touche2020 | 0.5126 | 0.5079 | 0.5110 | 0.5118 | 0.5118 | 0.5258 |
| AskUbuntuDupQuestions | 0.5605 | 0.5591 | 0.5622 | 0.5560 | 0.5611 | 0.5582 |
| MindSmallRerank | 0.3213 | 0.3202 | 0.3198 | 0.3208 | 0.3202 | 0.3182 |
| SciDocsRR | 0.6956 | 0.6964 | 0.6926 | 0.6967 | 0.6977 | 0.6977 |
| StackOverflowDupQuestions | 0.4176 | 0.4185 | 0.4163 | 0.4165 | 0.4194 | 0.4259 |
| BIOSSES | 0.7332 | 0.7400 | 0.7408 | 0.7530 | 0.7588 | 0.7477 |
| STS12 | 0.6337 | 0.6294 | 0.6282 | 0.6244 | 0.6290 | 0.6342 |
| STS13 | 0.7732 | 0.7731 | 0.7714 | 0.7698 | 0.7769 | 0.7730 |
| STS14 | 0.6745 | 0.6748 | 0.6741 | 0.6676 | 0.6764 | 0.6768 |
| STS15 | 0.8011 | 0.7976 | 0.7968 | 0.7951 | 0.7948 | 0.7971 |
| STS16 | 0.7741 | 0.7763 | 0.7755 | 0.7644 | 0.7694 | 0.7724 |
| STS17 | 0.7948 | 0.7965 | 0.7787 | 0.7914 | 0.7812 | 0.7925 |
| STS22 | 0.6790 | 0.6859 | 0.6785 | 0.6844 | 0.6828 | 0.6790 |
| AmazonCounterfactual | 0.5427 | 0.5385 | 0.5388 | 0.5981 | 0.6191 | 0.6002 |
| AmazonPolarity | 0.6072 | 0.5991 | 0.6030 | 0.6004 | 0.6048 | 0.5931 |
| AmazonReviews | 0.2787 | 0.2761 | 0.2841 | 0.2809 | 0.2845 | 0.2823 |
| Banking77 | 0.7237 | 0.7199 | 0.7232 | 0.7207 | 0.7240 | 0.7267 |
| Imdb | 0.5937 | 0.5932 | 0.5939 | 0.5757 | 0.5820 | 0.5762 |
| ToxicConversations | 0.5371 | 0.5611 | 0.5723 | 0.6117 | 0.5885 | 0.5979 |
| TweetSentimentExtraction | 0.4266 | 0.4176 | 0.4293 | 0.4344 | 0.4297 | 0.4238 |

Table 12: Full Benchmark Results for NomicBERT

| Dataset | (1,768) | (3,256) | (6,128) | (12,64) | (24,32) | (48,16) |
|---|---|---|---|---|---|---|
| ArguAna | 0.5795 | 0.5685 | 0.5944 | 0.6109 | 0.5997 | 0.5974 |
| ClimateFEVER | 0.3034 | 0.2915 | 0.2894 | 0.2947 | 0.2923 | 0.3059 |
| DBPEDIA | 0.5305 | 0.5398 | 0.5480 | 0.5399 | 0.5334 | 0.5482 |
| FEVER | 0.7814 | 0.8017 | 0.8181 | 0.8000 | 0.8120 | 0.8068 |
| FiQA2018 | 0.4463 | 0.4573 | 0.4556 | 0.4580 | 0.4472 | 0.4403 |
| HotpotQA | 0.6766 | 0.6631 | 0.6628 | 0.6741 | 0.6950 | 0.6950 |
| MSMARCO | 0.5748 | 0.5860 | 0.5840 | 0.6115 | 0.5863 | 0.5925 |
| NFCorpus | 0.2303 | 0.2229 | 0.2192 | 0.2219 | 0.2291 | 0.2329 |
| QuoraRetrieval | 0.8992 | 0.9095 | 0.9258 | 0.9143 | 0.9212 | 0.9342 |
| SCIDOCS | 0.2724 | 0.2630 | 0.2507 | 0.2638 | 0.2690 | 0.2657 |
| SciFact | 0.5880 | 0.5865 | 0.5958 | 0.5881 | 0.5876 | 0.5741 |
| Touche2020 | 0.5429 | 0.5251 | 0.5282 | 0.5286 | 0.5253 | 0.5299 |
| AskUbuntuDupQuestions | 0.5505 | 0.5525 | 0.5501 | 0.5521 | 0.5508 | 0.5516 |
| MindSmallRerank | 0.3052 | 0.3041 | 0.3062 | 0.3055 | 0.3053 | 0.3048 |
| SciDocsRR | 0.7085 | 0.7070 | 0.7068 | 0.7085 | 0.7114 | 0.7126 |
| StackOverflowDupQuestions | 0.4246 | 0.4262 | 0.4168 | 0.4181 | 0.4169 | 0.4188 |
| BIOSSES | 0.7975 | 0.7939 | 0.7922 | 0.7987 | 0.8063 | 0.8091 |
| STS12 | 0.6368 | 0.6375 | 0.6288 | 0.6280 | 0.6329 | 0.6393 |
| STS13 | 0.7812 | 0.7794 | 0.7723 | 0.7616 | 0.7656 | 0.7723 |
| STS14 | 0.7051 | 0.6974 | 0.6929 | 0.6919 | 0.6965 | 0.6994 |
| STS15 | 0.7951 | 0.7964 | 0.7931 | 0.7898 | 0.7918 | 0.7929 |
| STS16 | 0.7721 | 0.7677 | 0.7669 | 0.7739 | 0.7764 | 0.7772 |
| STS17 | 0.7941 | 0.8037 | 0.7878 | 0.7908 | 0.7960 | 0.7974 |
| STS22 | 0.6651 | 0.6607 | 0.6598 | 0.6637 | 0.6647 | 0.6635 |
| AmazonCounterfactual | 0.4996 | 0.5010 | 0.5043 | 0.5134 | 0.5231 | 0.5105 |
| AmazonPolarity | 0.5834 | 0.5857 | 0.5808 | 0.5899 | 0.5932 | 0.5934 |
| AmazonReviews | 0.2683 | 0.2682 | 0.2702 | 0.2733 | 0.2737 | 0.2740 |
| Banking77 | 0.7002 | 0.6996 | 0.7009 | 0.7046 | 0.7072 | 0.7063 |
| Imdb | 0.5579 | 0.5587 | 0.5630 | 0.5618 | 0.5622 | 0.5610 |
| ToxicConversations | 0.4737 | 0.4845 | 0.4879 | 0.5224 | 0.5269 | 0.5282 |
| TweetSentimentExtraction | 0.4203 | 0.4184 | 0.4188 | 0.4223 | 0.4148 | 0.4207 |

Table 13: Full Benchmark Results for ModernBERT

| Dataset | (1,768) | (3,256) | (6,128) | (12,64) | (24,32) | (48,16) |
|---|---|---|---|---|---|---|
| ArguAna | 0.5932 | 0.5979 | 0.5802 | 0.5852 | 0.5950 | 0.5937 |
| ClimateFEVER | 0.2782 | 0.2682 | 0.2567 | 0.1862 | 0.2969 | 0.2994 |
| DBPEDIA | 0.4932 | 0.5012 | 0.4954 | 0.5143 | 0.5108 | 0.5100 |
| FEVER | 0.7746 | 0.7867 | 0.8071 | 0.7306 | 0.7968 | 0.8051 |
| FiQA2018 | 0.4657 | 0.4711 | 0.4703 | 0.4734 | 0.4851 | 0.4726 |
| HotpotQA | 0.6561 | 0.6489 | 0.6505 | 0.6582 | 0.6612 | 0.6776 |
| MSMARCO | 0.5797 | 0.5680 | 0.5803 | 0.5713 | 0.6027 | 0.5755 |
| NFCorpus | 0.2263 | 0.2325 | 0.2248 | 0.2208 | 0.2387 | 0.2544 |
| QuoraRetrieval | 0.9450 | 0.9521 | 0.9405 | 0.9494 | 0.9552 | 0.9408 |
| SCIDOCS | 0.2979 | 0.3027 | 0.2982 | 0.2941 | 0.2849 | 0.2911 |
| SciFact | 0.6228 | 0.6040 | 0.6041 | 0.5936 | 0.6558 | 0.6640 |
| Touche2020 | 0.5279 | 0.5081 | 0.5179 | 0.5114 | 0.5146 | 0.5201 |
| AskUbuntuDupQuestions | 0.5895 | 0.5909 | 0.5891 | 0.5855 | 0.5849 | 0.5892 |
| MindSmallRerank | 0.3100 | 0.3133 | 0.3122 | 0.3171 | 0.3143 | 0.3154 |
| SciDocsRR | 0.7290 | 0.7320 | 0.7323 | 0.7319 | 0.7396 | 0.7406 |
| StackOverflowDupQuestions | 0.4756 | 0.4712 | 0.4733 | 0.4692 | 0.4746 | 0.4760 |
| BIOSSES | 0.8106 | 0.8125 | 0.8187 | 0.8132 | 0.8076 | 0.8103 |
| STS12 | 0.6480 | 0.6448 | 0.6554 | 0.6399 | 0.6534 | 0.6480 |
| STS13 | 0.7997 | 0.7888 | 0.7953 | 0.7710 | 0.7851 | 0.7834 |
| STS14 | 0.7203 | 0.7106 | 0.7162 | 0.7051 | 0.7109 | 0.7059 |
| STS15 | 0.8057 | 0.8044 | 0.8037 | 0.8017 | 0.8026 | 0.8033 |
| STS16 | 0.7687 | 0.7643 | 0.7609 | 0.7648 | 0.7745 | 0.7784 |
| STS17 | 0.7926 | 0.7981 | 0.7969 | 0.7983 | 0.7820 | 0.7816 |
| STS22 | 0.6615 | 0.6686 | 0.6689 | 0.6712 | 0.6718 | 0.6652 |
| AmazonCounterfactual | 0.5296 | 0.5333 | 0.5371 | 0.5468 | 0.5694 | 0.5686 |
| AmazonPolarity | 0.5875 | 0.5883 | 0.5883 | 0.5785 | 0.5895 | 0.5970 |
| AmazonReviews | 0.2775 | 0.2717 | 0.2719 | 0.2675 | 0.2765 | 0.2803 |
| Banking77 | 0.7587 | 0.7471 | 0.7504 | 0.7501 | 0.7509 | 0.7537 |
| Imdb | 0.5822 | 0.5863 | 0.5832 | 0.5754 | 0.5951 | 0.5953 |
| ToxicConversations | 0.4737 | 0.4807 | 0.4809 | 0.5165 | 0.5126 | 0.5281 |
| TweetSentimentExtraction | 0.4090 | 0.4094 | 0.4051 | 0.4156 | 0.4168 | 0.4148 |

Table 14: Full Benchmark Results for GTE-en-MLM

| Dataset | (1,768) | (3,256) | (6,128) | (12,64) | (24,32) | (48,16) |
|---|---|---|---|---|---|---|
| ArguAna | 0.5958 | 0.5943 | 0.5989 | 0.6015 | 0.6040 | 0.5975 |
| ClimateFEVER | 0.2631 | 0.2710 | 0.2408 | 0.2756 | 0.2802 | 0.2846 |
| DBPEDIA | 0.5206 | 0.5248 | 0.5260 | 0.5127 | 0.5143 | 0.5271 |
| FEVER | 0.7952 | 0.7885 | 0.7445 | 0.7837 | 0.7900 | 0.7809 |
| FiQA2018 | 0.4386 | 0.4685 | 0.4699 | 0.4812 | 0.4835 | 0.4799 |
| HotpotQA | 0.6389 | 0.6516 | 0.6440 | 0.6606 | 0.6618 | 0.6655 |
| MSMARCO | 0.6039 | 0.5871 | 0.6129 | 0.6220 | 0.6177 | 0.6099 |
| NFCorpus | 0.2884 | 0.2802 | 0.2866 | 0.2871 | 0.2839 | 0.2834 |
| QuoraRetrieval | 0.9236 | 0.9172 | 0.9134 | 0.9214 | 0.9197 | 0.9264 |
| SCIDOCS | 0.2842 | 0.2893 | 0.2915 | 0.2996 | 0.2949 | 0.3023 |
| SciFact | 0.6226 | 0.6088 | 0.6150 | 0.6126 | 0.6284 | 0.6077 |
| Touche2020 | 0.5180 | 0.5252 | 0.5250 | 0.5322 | 0.5379 | 0.5361 |
| AskUbuntuDupQuestions | 0.5825 | 0.5790 | 0.5771 | 0.5762 | 0.5773 | 0.5771 |
| MindSmallRerank | 0.3155 | 0.3162 | 0.3172 | 0.3167 | 0.3169 | 0.3156 |
| SciDocsRR | 0.7304 | 0.7321 | 0.7287 | 0.7314 | 0.7324 | 0.7321 |
| StackOverflowDupQuestions | 0.4409 | 0.4382 | 0.4367 | 0.4396 | 0.4446 | 0.4446 |
| BIOSSES | 0.8198 | 0.8157 | 0.8139 | 0.8111 | 0.8145 | 0.8210 |
| STS12 | 0.6410 | 0.6424 | 0.6346 | 0.6411 | 0.6428 | 0.6437 |
| STS13 | 0.7897 | 0.7929 | 0.7893 | 0.7903 | 0.7907 | 0.7925 |
| STS14 | 0.7112 | 0.7135 | 0.7084 | 0.7126 | 0.7146 | 0.7158 |
| STS15 | 0.8049 | 0.8092 | 0.8057 | 0.8100 | 0.8110 | 0.8114 |
| STS16 | 0.7676 | 0.7696 | 0.7698 | 0.7778 | 0.7798 | 0.7806 |
| STS17 | 0.8074 | 0.8110 | 0.8021 | 0.8103 | 0.8114 | 0.8144 |
| STS22 | 0.6556 | 0.6569 | 0.6571 | 0.6598 | 0.6611 | 0.6603 |
| AmazonCounterfactual | 0.5009 | 0.4949 | 0.4954 | 0.4982 | 0.5046 | 0.4828 |
| AmazonPolarity | 0.6007 | 0.6020 | 0.6029 | 0.6060 | 0.6083 | 0.5943 |
| AmazonReviews | 0.2774 | 0.2781 | 0.2755 | 0.2839 | 0.2860 | 0.2843 |
| Banking77 | 0.7394 | 0.7404 | 0.7407 | 0.7453 | 0.7456 | 0.7445 |
| Imdb | 0.5663 | 0.5693 | 0.5743 | 0.5815 | 0.5801 | 0.5779 |
| ToxicConversations | 0.4664 | 0.4663 | 0.5046 | 0.5184 | 0.5174 | 0.5185 |
| TweetSentimentExtraction | 0.4277 | 0.4234 | 0.4223 | 0.4168 | 0.4121 | 0.4113 |

Table 15: Retrieval Benchmark Results for Average Pooling Methods

| | ModernBERT | | | | | |
|---|---|---|---|---|---|---|
| Dataset | (1,768) | (3,256) | (6,128) | (12,64) | (24,32) | (48,16) |
| ArguAna | 0.5976 | 0.6003 | 0.6136 | 0.6001 | 0.6095 | 0.5961 |
| ClimateFEVER | 0.2759 | 0.2945 | 0.2897 | 0.3080 | 0.3061 | 0.3152 |
| DBPEDIA | 0.5233 | 0.5202 | 0.5173 | 0.4930 | 0.5021 | 0.4912 |
| FEVER | 0.7949 | 0.7731 | 0.7847 | 0.7854 | 0.7887 | 0.7856 |
| FiQA2018 | 0.4406 | 0.4471 | 0.4525 | 0.4455 | 0.4884 | 0.4886 |
| HotpotQA | 0.6555 | 0.6560 | 0.6401 | 0.6608 | 0.6638 | 0.6614 |
| MSMARCO | 0.5380 | 0.5266 | 0.5556 | 0.5255 | 0.5494 | 0.5469 |
| NFCorpus | 0.2502 | 0.2864 | 0.2658 | 0.2478 | 0.2666 | 0.2532 |
| QuoraRetrieval | 0.9518 | 0.9357 | 0.9361 | 0.9233 | 0.9550 | 0.9313 |
| SCIDOCS | 0.2922 | 0.2949 | 0.3060 | 0.2832 | 0.2958 | 0.2909 |
| SciFact | 0.6278 | 0.6556 | 0.6322 | 0.6352 | 0.6285 | 0.6534 |
| Touche2020 | 0.5123 | 0.5036 | 0.5005 | 0.4953 | 0.5118 | 0.5147 |

| | BERT | | | | | |
|---|---|---|---|---|---|---|
| Dataset | (1,768) | (3,256) | (6,128) | (12,64) | (24,32) | (48,16) |
| ArguAna | 0.5951 | 0.5971 | 0.5899 | 0.6012 | 0.5939 | 0.6004 |
| ClimateFEVER | 0.2527 | 0.2536 | 0.2598 | 0.2637 | 0.2461 | 0.2470 |
| DBPEDIA | 0.5091 | 0.5162 | 0.5184 | 0.5248 | 0.5271 | 0.5136 |
| FEVER | 0.8118 | 0.8108 | 0.8219 | 0.8272 | 0.8276 | 0.8382 |
| FiQA2018 | 0.3796 | 0.3703 | 0.3708 | 0.3797 | 0.3766 | 0.3653 |
| HotpotQA | 0.6386 | 0.6484 | 0.6424 | 0.6431 | 0.6428 | 0.6384 |
| MSMARCO | 0.5651 | 0.5783 | 0.5561 | 0.5531 | 0.5451 | 0.5662 |
| NFCorpus | 0.2572 | 0.2438 | 0.2495 | 0.2495 | 0.2473 | 0.2558 |
| QuoraRetrieval | 0.9303 | 0.9329 | 0.9230 | 0.9305 | 0.9306 | 0.9379 |
| SCIDOCS | 0.2481 | 0.2464 | 0.2479 | 0.2554 | 0.2537 | 0.2525 |
| SciFact | 0.5841 | 0.5858 | 0.5801 | 0.5802 | 0.5701 | 0.5732 |
| Touche2020 | 0.4695 | 0.4693 | 0.4663 | 0.4618 | 0.4641 | 0.4662 |

Table 16: Retrieval Benchmark Results for Fragment Granularity Scaling

| ModernBERT | | | | | | | | | | | |
|---|---|---|---|---|---|---|---|---|---|---|---|
| Dataset | (1, 768) | (2, 384) | (3, 256) | (4, 192) | (6, 128) | (12, 64) | (24, 32) | (48, 16) | (96, 8) | (192, 4) | (384, 2) |
| ArguAna | 0.5932 | 0.5853 | 0.5979 | 0.5890 | 0.5802 | 0.5852 | 0.5950 | 0.5937 | 0.6031 | 0.5885 | 0.5743 |
| ClimateFEVER | 0.2782 | 0.2617 | 0.2682 | 0.2608 | 0.2567 | 0.1862 | 0.2969 | 0.2994 | 0.3035 | 0.2920 | 0.2731 |
| DBPEDIA | 0.4932 | 0.4939 | 0.5012 | 0.4962 | 0.4954 | 0.5143 | 0.5108 | 0.5100 | 0.5081 | 0.5030 | 0.4900 |
| FEVER | 0.7746 | 0.8147 | 0.7867 | 0.7878 | 0.8071 | 0.7306 | 0.7968 | 0.8051 | 0.8079 | 0.7846 | 0.7619 |
| FiQA2018 | 0.4657 | 0.4689 | 0.4711 | 0.4835 | 0.4703 | 0.4734 | 0.4851 | 0.4726 | 0.4789 | 0.4668 | 0.4400 |
| HotpotQA | 0.6561 | 0.6482 | 0.6489 | 0.6435 | 0.6505 | 0.6582 | 0.6612 | 0.6776 | 0.6700 | 0.6540 | 0.6188 |
| MSMARCO | 0.5797 | 0.5976 | 0.5680 | 0.6046 | 0.5803 | 0.5713 | 0.6027 | 0.5755 | 0.5702 | 0.5501 | 0.5400 |
| NFCorpus | 0.2263 | 0.2237 | 0.2325 | 0.2241 | 0.2248 | 0.2208 | 0.2387 | 0.2544 | 0.2391 | 0.2318 | 0.2298 |
| QuoraRetrieval | 0.9450 | 0.9562 | 0.9521 | 0.9544 | 0.9405 | 0.9494 | 0.9552 | 0.9408 | 0.9676 | 0.9544 | 0.9588 |
| SCIDOCS | 0.2979 | 0.2881 | 0.3027 | 0.2869 | 0.2982 | 0.2941 | 0.2849 | 0.2911 | 0.2848 | 0.2654 | 0.2578 |
| SciFact | 0.6228 | 0.6005 | 0.6040 | 0.6055 | 0.6041 | 0.5936 | 0.6558 | 0.6640 | 0.6570 | 0.6487 | 0.6064 |
| Touche2020 | 0.5279 | 0.5175 | 0.5081 | 0.5243 | 0.5179 | 0.5114 | 0.5146 | 0.5201 | 0.5171 | 0.5183 | 0.5404 |

| BERT | | | | | | | | | | | |
|---|---|---|---|---|---|---|---|---|---|---|---|
| Dataset | (1, 768) | (2, 384) | (3, 256) | (4, 192) | (6, 128) | (12, 64) | (24, 32) | (48, 16) | (96, 8) | (192, 4) | (384, 2) |
| ArguAna | 0.5780 | 0.5841 | 0.5757 | 0.5724 | 0.5873 | 0.5837 | 0.5963 | 0.5736 | 0.5717 | 0.5847 | 0.5490 |
| ClimateFEVER | 0.2224 | 0.2285 | 0.2273 | 0.2399 | 0.2290 | 0.2243 | 0.2266 | 0.2181 | 0.2221 | 0.2052 | 0.2285 |
| DBPEDIA | 0.5148 | 0.5108 | 0.5129 | 0.5142 | 0.5132 | 0.5089 | 0.5168 | 0.5155 | 0.5208 | 0.5177 | 0.5099 |
| FEVER | 0.8221 | 0.8133 | 0.8034 | 0.8161 | 0.8208 | 0.8172 | 0.8253 | 0.8306 | 0.8212 | 0.8009 | 0.8504 |
| FiQA2018 | 0.3847 | 0.3879 | 0.3730 | 0.3725 | 0.3810 | 0.3862 | 0.3788 | 0.3774 | 0.3798 | 0.3640 | 0.3367 |
| HotpotQA | 0.6313 | 0.6462 | 0.6336 | 0.6474 | 0.6364 | 0.6222 | 0.6263 | 0.6330 | 0.6258 | 0.6210 | 0.6059 |
| MSMARCO | 0.5693 | 0.5703 | 0.5595 | 0.5685 | 0.5609 | 0.5503 | 0.5721 | 0.5661 | 0.5938 | 0.5709 | 0.5378 |
| NFCorpus | 0.2542 | 0.2652 | 0.2627 | 0.2610 | 0.2628 | 0.2679 | 0.2646 | 0.2690 | 0.2605 | 0.2610 | 0.2540 |
| QuoraRetrieval | 0.9267 | 0.9175 | 0.9208 | 0.9192 | 0.9210 | 0.9199 | 0.9193 | 0.9203 | 0.9206 | 0.9300 | 0.9293 |
| SCIDOCS | 0.2475 | 0.2427 | 0.2446 | 0.2362 | 0.2437 | 0.2468 | 0.2458 | 0.2487 | 0.2540 | 0.2409 | 0.2463 |
| SciFact | 0.5675 | 0.5740 | 0.5644 | 0.5722 | 0.5685 | 0.5648 | 0.5651 | 0.5571 | 0.5822 | 0.5692 | 0.4953 |
| Touche2020 | 0.4695 | 0.4726 | 0.4615 | 0.4689 | 0.4688 | 0.4665 | 0.4679 | 0.4739 | 0.4638 | 0.4599 | 0.4729 |

Table 17: Retrieval Benchmark Results for Decoder Models

| Qwen3-0.6B | | | | | | | |
|---|---|---|---|---|---|---|---|
| Dataset | (1, 2048) | (2, 1024) | (4, 512) | (8, 256) | (16, 128) | (32, 64) | (64, 32) | (128, 16) |
| ArguAna | 0.5885 | 0.6229 | 0.6168 | 0.6039 | 0.6081 | 0.6289 | 0.6214 | 0.6221 |
| ClimateFEVER | 0.3585 | 0.3471 | 0.3592 | 0.3488 | 0.3341 | 0.3686 | 0.3524 | 0.3148 |
| DBPEDIA | 0.5104 | 0.5084 | 0.5371 | 0.5004 | 0.5477 | 0.5416 | 0.5305 | 0.5524 |
| FEVER | 0.8833 | 0.8796 | 0.8293 | 0.8745 | 0.8398 | 0.8627 | 0.8586 | 0.8196 |
| FiQA2018 | 0.4891 | 0.4889 | 0.5152 | 0.4904 | 0.5180 | 0.5193 | 0.5160 | 0.4853 |
| HotpotQA | 0.7246 | 0.7322 | 0.7216 | 0.7176 | 0.7384 | 0.7088 | 0.7413 | 0.7248 |
| MSMARCO | 0.5260 | 0.5643 | 0.6162 | 0.5666 | 0.5790 | 0.5945 | 0.5784 | 0.6013 |
| NFCorpus | 0.2719 | 0.2652 | 0.2897 | 0.2881 | 0.2898 | 0.2798 | 0.2855 | 0.3017 |
| QuoraRetrieval | 0.9414 | 0.9409 | 0.9397 | 0.9405 | 0.9396 | 0.9429 | 0.9450 | 0.9470 |
| SCIDOCS | 0.3770 | 0.3717 | 0.3735 | 0.3774 | 0.3774 | 0.3803 | 0.3701 | 0.3829 |
| SciFact | 0.7464 | 0.7441 | 0.7366 | 0.7251 | 0.7333 | 0.7511 | 0.7520 | 0.7659 |
| Touche2020 | 0.5146 | 0.5031 | 0.5276 | 0.5375 | 0.5066 | 0.5159 | 0.5208 | 0.5410 |

| Llama-3.2-1B | | | | | | | |
|---|---|---|---|---|---|---|---|
| Dataset | (1, 2048) | (2, 1024) | (4, 512) | (8, 256) | (16, 128) | (32, 64) | (64, 32) | (128, 16) |
| ArguAna | 0.6451 | 0.6215 | 0.6365 | 0.6276 | 0.6407 | 0.6349 | 0.6320 | 0.6546 |
| ClimateFEVER | 0.3129 | 0.3086 | 0.3043 | 0.3208 | 0.3142 | 0.3194 | 0.3278 | 0.3183 |
| DBPEDIA | 0.5191 | 0.5514 | 0.5425 | 0.5589 | 0.5550 | 0.5344 | 0.5445 | 0.5590 |
| FEVER | 0.8917 | 0.9214 | 0.8981 | 0.9095 | 0.8804 | 0.8720 | 0.8583 | 0.8919 |
| FiQA2018 | 0.5454 | 0.5374 | 0.5368 | 0.5391 | 0.5450 | 0.5205 | 0.5448 | 0.5165 |
| HotpotQA | 0.7963 | 0.7843 | 0.7740 | 0.7764 | 0.7683 | 0.7944 | 0.7817 | 0.7810 |
| MSMARCO | 0.5589 | 0.5831 | 0.5961 | 0.5858 | 0.5965 | 0.6122 | 0.6289 | 0.5985 |
| NFCorpus | 0.3253 | 0.3463 | 0.3298 | 0.3338 | 0.3180 | 0.3345 | 0.3350 | 0.3354 |
| QuoraRetrieval | 0.9387 | 0.9483 | 0.9470 | 0.9505 | 0.9506 | 0.9351 | 0.9433 | 0.9441 |
| SCIDOCS | 0.3712 | 0.3592 | 0.3709 | 0.3719 | 0.3646 | 0.3642 | 0.3554 | 0.3672 |
| SciFact | 0.7324 | 0.7422 | 0.7563 | 0.7333 | 0.7380 | 0.7287 | 0.7243 | 0.7532 |
| Touche2020 | 0.4973 | 0.5170 | 0.5093 | 0.5131 | 0.5089 | 0.5065 | 0.5151 | 0.5076 |

Table 18: Retrieval Benchmark Results for ModernBERT with Different Similarity Metrics

| Dataset | Cosine | | Euclidean | | Manhattan | | Dot | |
|---|---|---|---|---|---|---|---|---|
| | (1, 768) | (48, 16) | (1, 768) | (48, 16) | (1, 768) | (48, 16) | (1, 768) | (48, 16) |
| ArguAna | 0.5932 | 0.5937 | 0.4801 | 0.5434 | 0.5060 | 0.5886 | 0.4800 | 0.5597 |
| ClimateFEVER | 0.2782 | 0.2994 | 0.0492 | 0.2157 | 0.0465 | 0.2150 | 0.1143 | 0.2419 |
| DBPEDIA | 0.4932 | 0.5100 | 0.2358 | 0.3313 | 0.2963 | 0.5163 | 0.2293 | 0.4879 |
| FEVER | 0.7746 | 0.8051 | 0.0885 | 0.7061 | 0.1142 | 0.7159 | 0.5152 | 0.7304 |
| FiQA2018 | 0.4657 | 0.4726 | 0.2764 | 0.3646 | 0.2997 | 0.4548 | 0.2037 | 0.4403 |
| HotpotQA | 0.6561 | 0.6776 | 0.6312 | 0.6576 | 0.6316 | 0.6218 | 0.6847 | 0.7034 |
| MSMARCO | 0.5797 | 0.5755 | 0.3738 | 0.4660 | 0.4001 | 0.5457 | 0.2653 | 0.5287 |
| NFCorpus | 0.2263 | 0.2544 | 0.1179 | 0.1647 | 0.1464 | 0.2030 | 0.1135 | 0.2014 |
| QuoraRetrieval | 0.9450 | 0.9408 | 0.9259 | 0.9338 | 0.9288 | 0.9475 | 0.8081 | 0.9536 |
| SCIDOCS | 0.2979 | 0.2911 | 0.2080 | 0.2374 | 0.2667 | 0.2619 | 0.1503 | 0.2706 |
| SciFact | 0.6228 | 0.6640 | 0.4414 | 0.5955 | 0.4716 | 0.5986 | 0.4076 | 0.6039 |
| Touche2020 | 0.5279 | 0.5201 | 0.3907 | 0.4668 | 0.4196 | 0.5021 | 0.3310 | 0.4937 |

Table 19: Retrieval Benchmark Results on ModernBERT: MRL vs. SFS (trained with $d_{frag} = 16$)

| | MRL | | | | | |
|---|---|---|---|---|---|---|
| Dataset | 64 | 128 | 256 | 384 | 512 | 768 |
| ArguAna | 0.5466 | 0.5405 | 0.5565 | 0.5654 | 0.5695 | 0.5757 |
| ClimateFEVER | 0.1988 | 0.2500 | 0.2678 | 0.2578 | 0.2663 | 0.2883 |
| DBPEDIA | 0.4465 | 0.4457 | 0.4773 | 0.5019 | 0.5048 | 0.5108 |
| FEVER | 0.6840 | 0.6898 | 0.7248 | 0.7928 | 0.7784 | 0.7941 |
| FiQA2018 | 0.3174 | 0.3484 | 0.4274 | 0.4266 | 0.4484 | 0.4603 |
| HotpotQA | 0.5407 | 0.5780 | 0.6066 | 0.6050 | 0.6402 | 0.6278 |
| MSMARCO | 0.5313 | 0.5342 | 0.5768 | 0.5613 | 0.5452 | 0.5612 |
| NFCorpus | 0.1627 | 0.1900 | 0.2050 | 0.2100 | 0.2109 | 0.2236 |
| NQ | 0.4743 | 0.4984 | 0.5357 | 0.5301 | 0.5238 | 0.5230 |
| QuoraRetrieval | 0.9286 | 0.9462 | 0.9441 | 0.9548 | 0.9479 | 0.9478 |
| SCIDOCS | 0.2085 | 0.2414 | 0.2544 | 0.2897 | 0.2877 | 0.2891 |
| SciFact | 0.4845 | 0.5479 | 0.5692 | 0.6103 | 0.6043 | 0.5958 |
| Touche2020 | 0.4701 | 0.4858 | 0.4890 | 0.4955 | 0.5091 | 0.5161 |
| | SFS | | | | | |
| Dataset | 64 (4) | 128 (8) | 256 (16) | 384 (24) | 512 (36) | 768 (48) |
| ArguAna | 0.5015 | 0.5156 | 0.5747 | 0.5963 | 0.5953 | 0.5937 |
| ClimateFEVER | 0.2023 | 0.2347 | 0.3020 | 0.2941 | 0.3113 | 0.2994 |
| DBPEDIA | 0.3676 | 0.4268 | 0.4677 | 0.4916 | 0.4926 | 0.5100 |
| FEVER | 0.5891 | 0.7333 | 0.7919 | 0.7728 | 0.7990 | 0.8051 |
| FiQA2018 | 0.3353 | 0.3756 | 0.4340 | 0.4606 | 0.4795 | 0.4726 |
| HotpotQA | 0.5042 | 0.5840 | 0.6196 | 0.6525 | 0.6850 | 0.6776 |
| MSMARCO | 0.5071 | 0.5264 | 0.5761 | 0.5670 | 0.5802 | 0.5755 |
| NFCorpus | 0.1226 | 0.1445 | 0.2193 | 0.2467 | 0.2338 | 0.2544 |
| NQ | 0.3836 | 0.4491 | 0.5341 | 0.5408 | 0.5728 | 0.5616 |
| QuoraRetrieval | 0.9181 | 0.9167 | 0.9357 | 0.9447 | 0.9460 | 0.9408 |
| SCIDOCS | 0.1633 | 0.1961 | 0.2488 | 0.2832 | 0.2956 | 0.2911 |
| SciFact | 0.5114 | 0.5646 | 0.5954 | 0.6354 | 0.6355 | 0.6640 |
| Touche2020 | 0.4530 | 0.4729 | 0.4977 | 0.5038 | 0.5068 | 0.5201 |

