# OpenReview forum: "Semantic Fragment Similarity Representation Learning for Information Retrieval"
_ICLR.cc/2026/Conference — ICLR 2026 Conference Withdrawn Submission_

### Official Review · Reviewer_V7vt · 2025-10-25

**Soundness:** 2
**Presentation:** 3
**Contribution:** 2
**Rating:** 4
**Confidence:** 4

**Summary:**

The authors propose a Semantic Fragment similarity where they propose to split an embedding dimension into multiple equi-dimensional fragments and train the fragements. they observe some improvements in different tasks like retrieval, classification etc. and also look at the difference between the features (mesaured as sd per fragment of embedding differences for classes).

**Strengths:**

1. The work tackles an important problem of sematic structure loss due to embedding pooling during dense vector retrieval. This is an important aspect to study from research and practice.
2. Their comprehesive experiments establish some improvements across tasks.
3. The improvements with respect to MRL is interesting.

**Weaknesses:**

1. The authors proposed approach is not very dissimilar to the concept of subspace embeddings. A clearer separation of their contribution from the literature would have been illuminating. [1],[2] are examples from a quick search. A more focussed search can bring about more results.

2.  One aspect which comes up when going through the authors formulation is why have they chosen to split the spaces by dimensions and not taken an approach more aligned to PCA / SVD analysis. There is no discussion on this.

3. This brings us to the most important aspect of the subspace formulation. Their equation (4) is not very different from training multiple embeddings - the separation of the d-dimensional space into N_f fragments would mean that we would want the fragments to learn different semantic aspects and so we would like to bring in some form of orthogonality (even if not strictly orthogonal). Unless I am missing something there is no part in the training which looks at the interrelation (or the lack of it) between fragements.

4. The numbers in Tables 1-4 have not been analyzed for statistical significance. For example in Table 1 the best scores in bold differ from the baseline score in the first column (1 fragment) often by less than 1 percentage point (0.5197 vs 0.5178). Such small score improvements need significance testing for reliability. Similar observations on Table 2 also.

5. "Relative to the baseline, Toxic Conversation increases by up to 11.48%, and Amazon Counterfactual by up to 7.51%." - I am observing 52.81 vs 47.27 and 56.94 vs 52.96. Not sure what I am missing to not notice the claimed accuracy improvements

6. I am not clear on why figure 3a (also missing statistical significance testing of actual differences in the distributions) should be a measure of semantic independence of fragments.

7. cosine similarity in Figure 3b should be used with caution [3,4](also other works) for intepreting similarity when comparing across models and should only be used for intra-model ranks.

8. "In other words, using only 16 dimensions (about 12% of 768) can outperform the full embedding, suggesting that training with fragment similarity concentrates semantic signals within fragments." - is there an option to therefore have smaller models learnt in this way? One aspect of future research could also be trying to see if a subset of such fragements are sufficient for performance. Many resource constrained systems would benefit from this.

9. lines 178 - 182 - is this training from scratch or fine tuning / continued pre-training. This detail is missing. If the latter what is the effect of the individual corresponding subspaces i.e. if the original model was split up into subspaces based on same defrag would we have similar results.

One important aspect which would have been illuminating is experiments to see which conceptual aspects are captured by which segment by comparing the performance across different types of queries/datasets.

Minor comments.

1. The variables in the equations especially equation 4 should be clearly defined. for example what is i and j - although one can make a fairly educated guess.


[1] Jegou, Herve, Matthijs Douze, and Cordelia Schmid. "Product quantization for nearest neighbor search." IEEE transactions on pattern analysis and machine intelligence 33.1 (2010): 117-128.

[2]Li, Zhi, et al. "Disentangling Latent Embeddings with Sparse Linear Concept Subspaces (SLiCS)." arXiv preprint arXiv:2508.20322 (2025).

[3] Steck, Harald, Chaitanya Ekanadham, and Nathan Kallus. "Is cosine-similarity of embeddings really about similarity?." Companion Proceedings of the ACM Web Conference 2024. 2024.

[4] S. Soman and S. Roychowdhury, “Observations on Building RAG Systems for Technical Documents,” in The Second Tiny Papers Track at ICLR 2024, Vienna, Austria, May 2024. [Online]. Available: https://openreview.net/forum?id=RFujq4HoV4

**Questions:**

1. Can the authors establish statistical significance of their results with the baseline results (without fragmentation) for the tables please.
2. Clarity on weaknees (3) and how is it addressed if relevant.

---

### Official Review · Reviewer_5gf6 · 2025-10-28

**Soundness:** 3
**Presentation:** 3
**Contribution:** 3
**Rating:** 6
**Confidence:** 3

**Summary:**

This paper introduces Semantic Fragment Similarity (SFS), a simple metric that replaces cosine similarity with an average of fragment-level similarities. The goal is to prevent the "flattening" of semantic features and encourage fragments to specialize. Empirically, SFS consistently improves performance across diverse models and MTEB tasks, and the analysis provides strong evidence for the claim of semantic specialization. The only major concern is that the SFS metric appears fundamentally incompatible with standard ANN libraries (e.g., FAISS). This makes it computationally infeasible for the paper's primary stated task of large-scale retrieval.

**Strengths:**

- The proposed method is very simple, requiring no changes to the model architecture and adding no new parameters. This makes it highly practical and easy to adopt.

- The authors have done a good job validating SFS. The method shows consistent gains across diverse models and diverse tasks.

- The experiment findings are insightful. For example,  the finding that a single, small fragment can outperform the entire embedding on a specific classification task might spark future research in this direction.

**Weaknesses:**

- Information retrieval at scale requires approximate nearest neighbor search (e.g., using FAISS). However, the proposed SFS metric seems incompatible with standard ANN methods. Also, algorithm 1 implies a brute-force scan over the entire corpus. This incompatibility limits the application of this method.

- The paper claims SFS opens avenues for interpretability. While the analysis in Section 6.1 shows that specialization occurs, it doesn’t show what the fragments learn.

**Questions:**

- How do the authors propose to use SFS for efficient, large-scale retrieval? Would this require N separate ANN indexes, one for each fragment, followed by a complex aggregation of scores?
- For interpretability, wonder if can "label" the semantic role of each fragment?

---

### Official Review · Reviewer_Howu · 2025-10-29

**Soundness:** 1
**Presentation:** 2
**Contribution:** 1
**Rating:** 2
**Confidence:** 4

**Summary:**

This paper proposed a fragmented way of computing the relevance score between a query vector and a document vector for retrieval. The authors proposed that the vector be segmented into multiple chunks, then a cosine similarity score is computed between each respective pairs of chunks, then taking an average. The authors claimed that this method results in good results for various IR benchmarks (e.g. BEIR) over some base models (e.g. BERT, RoBERTa, etc.; Qwen3-0.6B, Llama3.2-1B).

**Strengths:**

- The paper is written clearly, with the proposed similarity measure easy to understand
- Tested over a wide range of tasks and datasets

**Weaknesses:**

- The proposed method is not convincing. What it does is basically $L_2$-normalizing each chunk of the original vector and then compute an inner product. It is hard to see why this would perform better than a single vector.
 - The experiments also confirmed what I doubted: The improvements of chunking are marginal: the biggest gain are about 1%, and there are cases where the gain is about 0.05% over the baseline (chunk size 1). This is mostly statistically insignificant.
 - The inference algorithm is super slow as it need to compute the score over EACH document. Sparse retrieval like BM25 can utilize postings lists; and dense retrieval can use a wide range of approximate nearest neighbor algorithms. But this inference algorithm would not be practical in real-world scenarios.

**Questions:**

- Why is Qwen3-0.6B used, where there is an embedding model in the Qwen3 family?

---

### Official Review · Reviewer_qqfF · 2025-10-29

**Soundness:** 3
**Presentation:** 3
**Contribution:** 3
**Rating:** 4
**Confidence:** 4

**Summary:**

This paper proposes a novel similarity measurement metric, Semantic Fragment Similarity (SFS), to improve sentence representation quality. Traditional global vector-based cosine similarity flattens and entangles multi-faceted semantic features, which may be harmful for accuracy. The proposed SFS alleviates this issue by partitioning global vectors into non-overlapping fragments and computing fragment-level similarity.

**Strengths:**

- This paper is well written and very easy to follow.
- This paper addresses the flattening and entangling of multi-faceted semantic features caused by global vector cosine similarity. It is a good research point.
- The proposed idea of fragment similarity is novel and interesting.
- The experiments are extensive, covering multiple tasks and datasets.

**Weaknesses:**

- From the results on information retrieval datasets (Table 1, 3, 4), the improvement is not consistent for fragment similarity. Notably, Table 4 shows that most fragment similarity settings negatively impact performance compared to the base setting. Also in Table 2, the improvement for other tasks is not consistent. This contradicts the main claim. Given this evidence, I worry about its generalization ability and robustness, especially for retrieval tasks.
- Interestingly, in Table 5, fragment similarity consistently improves the decoder-only backones. However, it lacks a sufficient discussion of the performance discrepancy between the encoder and decoder backbones.
- Although it generally improves classification significantly (found in Table 7), this does not seem to be strong evidence to support the claim since this paper focuses on information retrieval tasks. Given this fact, I would suggest either focusing on classification tasks to gain more insights or providing more convincing results on retrieval tasks.
- How to find a proper fragment size/granularity is difficult and probably needs extra or extensive experiments for selection.
- I think the depth still can be improved; more discussions about how fragments work should be investigated. for example:
    - The current fragment similarity approach computes cosine similarity only between aligned fragment pairs (i.e., cos(q_i, d_i) where the i-th query fragment matches with the i-th document fragment). It would be valuable to explore cross-fragment similarity (i.e., cos(q_i, d_j)), similar to the late interaction mechanism in ColBERT [1]. This could capture richer semantic relationships between different aspects of queries and documents.
    - From the experiments on encoder similarity, SFS does not consistently outperform global similarity. It would be interesting to investigate how to combine them effectively. The global vector captures the overall semantics, while the fragment vectors capture local semantic differences.

**Questions:**

**Questions and Suggestions**

- How to apply the fragment for classification tasks? Does it also use the accumulated cosine similarity?
- How is its efficiency compared to global vector similarity in terms of computational cost?
- IIRC, this paper [2] breaks down global vectors into two parts and then performs operations in complex space to compute similarity. Are there similar methods that could be used in fragment similarity to improve its effectiveness for retrieval tasks?
- It would be better to discuss the computational overhead of fragment similarity compared to global similarity.


**Reference**:
- [1] Khattab O, Zaharia M. Colbert: Efficient and effective passage search via contextualized late interaction over bert[C]//Proceedings of the 43rd International ACM SIGIR conference on research and development in Information Retrieval. 2020: 39-48.
- [2] Li X, Li J. Angle-optimized text embeddings[J]. arXiv preprint arXiv:2309.12871, 2023.

---

### Note · Authors · 2025-12-03

I have read and agree with the venue's withdrawal policy on behalf of myself and my co-authors.